



# Statistical Study and Corresponding Evolution of Plasmaspheric Plumes under Different Levels of Geomagnetic Storms

Haimeng Li[1], Tongxing Fu[1], Rongxin Tang[1,2], Zhigang Yuan[3], Zhihai Ouyang[1], Xiaohua Deng[1]

[1] Institute of Space Science and Technology, Nanchang University, Nanchang, China.
[2] Jiangxi Provincial Key Laboratory of Interdisciplinary Science, Nanchang University, Nanchang, China.
[3] School of Electronic Information, Wuhan University, Wuhan, China

*Correspondence to*: Rongxin Tang, rongxint@ncu.edu.cn

**Abstract.** Using observations of Van Allen Probes, we present a statistical study of plasmaspheric plumes in the inner magnetosphere. Plasmaspheric plumes tend to occur during the recovery phase of geomagnetic storms. Furthermore, the results imply that the occurrence rate of observed plasmaspheric plume in the inner magnetosphere is larger during stronger geomagnetic activity. This statistical result is different from the observations of the Cluster satellite with much higher L-shells in most orbital period, which suggest that the plasmaspheric plume near the magnetopause tends to be observed during moderate geomagnetic activity (Lee et al., 2016). In the following, the dynamic evolutions of plasmaspheric plumes during a moderate geomagnetic storm in February 2013 and a strong geomagnetic storm in May 2013 are simulated through group test particle simulation. It is obvious that the plasmaspheric particles drift out on open convection paths due to sunward convection during both geomagnetic storms. It seems that the outer plasmaspheric particles exhaust sooner and the plasmasphere shrinks faster during strong geomagnetic storms. As a result, the longitudinal width of the plume is narrower and the plume is limited to lower L-shells during the recovery phase of strong geomagnetic storm. The simulated evolutions may provide a possible interpretation for the occurrence rates: Van Allen Probes tend to observe plumes during stronger geomagnetic storms, and the Cluster satellite with higher L-shells tends to observe plumes during moderate geomagnetic storms.

## 1 Introduction

The innermost magnetosphere is occupied by the torus of cold dense plasma known as the plasmasphere (Lemaire et al., 1998). In general, the dynamics of plasmaspheric particles are controlled by the combination of corotational and solar wind-driven convection electric fields. The southward interplanetary magnetic field (IMF) at the magnetopause brings about dayside magnetopause reconnection, resulting in an increase in dawn-dusk convection electric fields in the inner magnetosphere (Dungey, 1961). Goldstein et al. (2005a) suggested that the electric field at the plasmapause was approximately 13% of the solar wind electric field ($E_{SW}$). Under the effect of a dawn-dusk convection electric field, plasmaspheric particles move sunward through the $E \times B$ drift, and may transfer into the magnetospheric boundary layers. This dynamic mechanism leads to the erosion of the plasmasphere and the formation of a plasmaspheric plume near the dusk side (Goldstein et al., 2004; Darrouzet et al., 2008; Walsh et al., 2013). Long-term observations also suggest that the radial location of the plasmapause can move inward during periods of geomagnetic disturbance, which are mainly correlated with increases in the southward IMF (Elphic et al., 1996; Carpenter and Lemaire, 1997). After the time interval of the geomagnetic disturbance, low energy ionospheric particles are drawn upward from low altitudes along magnetic field lines, and contribute to the refilling of the eroded plasmasphere. It may take more than 10 days to recover to the normal level of the plasmasphere (Chu et al., 2017; Lointier et al., 2013).

The plasmaspheric plume is an important region of 'detached plasma elements' in the magnetosphere, it connects to the main body of the plasmasphere and stretches outward (Goldstein et al., 2004; Darrouzet et al., 2009b; Moldwin et al., 2016). Therefore, the plasmaspheric plume provides an effective coupling channel of energy/mass between the inner magnetospheric



plasmasphere and outer magnetosphere. During geomagnetic storms, the plasmaspheric plume may reach the dayside
magnetopause and thus reduce the reconnection rate (Dargent et al., 2020). Furthermore, structureless hiss waves and
electromagnetic ion cyclotron (EMIC) waves often arise in high-density plasmaspheric plumes (Meredith et al., 2004; Yuan
et al., 2012; Usanova et al., 2013; Grison et al., 2018; Yu et al., 2016; Yuan et al., 2010; Zhang et al., 2018, 2019). The electron
scattering induced by hiss waves is thought to be a key contributor to the formation of the radiation belt slot region (Su et al.,
2015; Shi et al., 2019; Zhang et al., 2019). Therefore, it is very important to study the formation and evolution of plasmaspheric
plumes. Generally, plasmaspheric plumes are identified when the electron density is more than the modeled density of the
plasmasphere (provided by Sheeley et al. (2001)) in a specific L-shell outside the plasmapause (Moldwin et al., 2004; Zhang
et al., 2019). Using density data from the Cluster spacecraft, Darrouzet et al. (2008) and Lee et al. (2016) presented statistical
studies of plasmaspheric plumes. Since the time interval of Cluster in the outer magnetosphere is much greater than that in the
inner magnetosphere, Cluster provides a good opportunity to investigate plumes in the outer magnetosphere. Studies suggest
that the occurrence rate of plasmaspheric plumes is significantly higher on the afternoon side than on the prenoon side, and
plasmaspheric plumes tend to be observed during moderate geomagnetic activity.
In this paper, data from Van Allen Probes are used in situ measurement is used to identify plasmaspheric plumes in the inner
magnetosphere (with L-shells $\leq$ 6). Plasmaspheric plume spatial distributions and occurrence rates at different levels of
geomagnetic activity are investigated. The results imply that the occurrence rate of plasmaspheric plumes in the inner
magnetosphere is largest during strongest geomagnetic activity, which is different from the statistical result near the
magnetopause provided by Lee et al. (2016). Moreover, to explain the different occurrence rates of observed plasmaspheric
plumes as a function of the levels of geomagnetic activity, group test particle simulations are used to exhibit the evolution of
plasmaspheric plumes during both moderate and strong geomagnetic activity.
**2 Data and Methodology**
In our study, using the observations of Van Allen Probe A, we performed statistical research on plasmaspheric plumes in the
inner magnetosphere. The perigee of Van Allen Probe is ~1.1 $R_E$ (radius of the Earth), and its apogee is ~6.2 $R_E$. Electron
density data with a 6.5 s time resolution are provided by Level 4 of the Electric and Magnetic Field Instrument Suite and
Integrated Science (EMFISIS) data sets of Van Allen Probe A (Kletzing et al., 2013), which is mainly calculated from the
trace of the upper hybrid resonance frequency (Kurth et al., 2015). Using electron density data, the structure of the
plasmaspheric plume is identified based on the following criteria. (1) The plasmapause is identified as the innermost steep
gradient of electron density, which requires the electron density to decrease by a factor >5 within 0.5 L-shell (Moldwin et al.,
2002; Malaspina et al., 2016; Zhang et al., 2019). Through the above criterion of the plasmapause, a very small number of
identified events are not the real plasmapause. To ensure the accuracy of the plasmapause database, these spurious events are
deleted artificially. (2) While Van Allen Probes are outside the plasmapause, we identify the region where the observed electron
density sharply increases, and the observed density exceeds the density calculated by the model of Sheeley et al. (2001) as
follow:
$$n_e = 1390 \left(\frac{3}{L}\right)^{4.83} - 240 \left(\frac{3}{L}\right)^{3.60} \tag{1}$$

Referencing the criterion of plasmaspheric plume identification in Darrouzet et al. (2008) and Zhang et al. (2019), if the satellite
orbit range of enhanced electron density is more than 0.2 $R_E$ and less than 2 $R_E$, we consider the region can be identified as a
plasmaspheric plume by satellite.
Figure 1 displays an example of a plasmaspheric plume observed by Van Allen Probe A from 06:30 UT to 13:20 UT on 6 June
2013. According to the criterion above, the location of the plasmapause is indicated by black vertical lines. While the satellite
is outside the plasmapause, the measured electron density (blue curve) from 07:25 UT to 08:10 UT (marked by gray shadow)



absolutely exceeds the density model provided by the Sheeley et al. (2001) model (red curve). As a result, the region of high
density marked by gray shadow is considered a plasmaspheric plume.
**3 Statistics of Observation**
Following the criterion method described above, we capture 422 plasmaspheric plume events out of 4030 Van Allen Probe A
orbits in the inner magnetosphere from January 2013 to December 2018. In this study, the global spatial distributions of
plasmaspheric plumes associated with different geomagnetic phases are analyzed. For a geomagnetic storm, the minimum Dst
must be at least below -30 nT, and the duration of that Dst ≤ -30 nT must be more than 10 minutes (Gonzalez et al., 1994).
The geomagnetic storm onset, which indicates the beginning of a geomagnetic storm, is defined as the time when the slope of
the Dst index becomes negative and remains negative until the minimum of Dst index. Then, 3 hours (hr) before the time of
onset is defined as the initial phase, as in Halford et al. (2010) and Wang et al. (2016). The period from the onset to the
minimum Dst in the geomagnetic storm is defined as the main phase, while the recovery phase begins after the minimum Dst
and ends when the Dst recovers to 80% of the minimum value or the next storm starts. The statistical outcome shows that 185
plasmaspheric plume events are detected during the nonstorm period. These events during the nonstorm period account for
43.8% percent of the total. The high proportion may be due to the relatively quiet geomagnetic activity during most of the time
interval. As shown in Figure 2a, it seems that the nonstorm plasmaspheric plume events cover all magnetic local time (MLT)
ranges. However, the maximum number of plasmaspheric plume events occurs from MLT~18 to MLT~24. The spatial
distributions of plasmaspheric plumes during different phases of geomagnetic storms are shown in Figure 2b-d. The numbers
of plasmaspheric plume events in the initial, main and recovery phases are 31, 32 and 174, respectively. During geomagnetic
storms, it seems that the plasmaspheric plume events observed in the recovery phase (174) occupy the largest proportion, and
the plasmaspheric plumes in the recovery phase are mainly located on the dusk side. On the other hand, the numbers of
plasmaspheric plumes in both the initial and main phases are lower (31 and 32, respectively). The plasmaspheric plumes in
the initial phase are mainly observed on the dusk-midnight side, and the plasmaspheric plumes in the main phase mainly occur
on the afternoon side.
Furthermore, we also examine the relationship between the occurrence rate of plasmaspheric plumes and the levels of
geomagnetic disturbance. Similar to the analysis of the relationship between the plasmaspheric plume near magnetopause and
geomagnetic activity studied in Lee et al. (2016), we selected the minimum Dst value from the previous 24 hr to account for
the response time of the plasmapause to geomagnetic activity, which was also adopted by Moldwin et al. (2004) and Darrouzet
et al. (2008). Figure 3a shows the distribution of observed plasmaspheric plume density data points as a function of minimum
Dst in the previous 24 hr. Notably, every density data point provided by Van Allen Probes during the interval of a plume event
is considered as one plasmaspheric plume sample. Figure 3b shows the normalized occurrence rates of plasmaspheric plumes
in the inner magnetosphere with respect to the minimum Dst in the previous 24 hr, which is obtained from the number of
density data points in the plasmaspheric plume divided by that of all density data points provided by Van Allen Probes during
the different levels of geomagnetic activity. It seems that the occurrence rates of plasmaspheric plumes in the interval of -10
< Dst < -10 nT are lower. On the other hand, the occurrence rates in intervals of -70 < Dst < -50 nT, -50 < Dst < -30 nT and -
30 < Dst < -10 nT are higher. The occurrence rates in the three intervals when -10 nT Dst < 10 nT are similar, but the occurrence
rate increases slightly with increasing geomagnetic activity level. The statistical results from Van Allen Probes are somewhat
different from the statistical result of plasmaspheric plumes near the dayside magnetopause measured by the Cluster spacecraft
displayed in Lee et al. (2016). The results of Lee et al. (2016) implied that plasmaspheric plumes near the magnetopause with
high L-shells tend to be observed during moderate geomagnetic activity, and the highest occurrence rate is in the interval -30
< Dst < -10 nT.



## 4 Simulated Evolution of plasmaspheric Plume

### 4.1 Model Inputs

Test particle simulation is a useful method to analyze the motions and variations in plasma (Zhou et al., 2018). To explain the disparity in the occurrence rates disparity of the observed plasmaspheric plume associated with geomagnetic activity levels in different L-shells (L ≤ 6.2 in the inner magnetosphere observed by the Van Allen Probe A satellite, and L ≥ 6.2 during most of the Cluster orbital period), we run a group test particle simulation to analyze the evolution of plasmaspheric plumes during different levels of geomagnetic storms. By calculating the drift paths of a great quantity of test plasmaspheric particles, the simulation not only provides the evolution of the plasmapause and plasmaspheric plume boundaries, which is similar to the plasmapause test particle (PTP) simulation provided by Goldstein et al. (2003, 2005a, b, 2014b), but also reveals the evolution of equatorial density in both the plasmasphere and plasmaspheric plume.

In this study, the geomagnetic field is assumed to be a dipolar field, and electron motion is assumed to be adiabatic. Following Goldstein et al. (2003, 2005a), we establish a magnetospheric model for the electric potential. The electric potential is the sum of the corotation electric potential $\Phi_{rot}$ and convection electric potential $\Phi_{VS}$:

$$\Phi_{rot} = -C\frac{R_E}{R} \tag{2}$$

$$\Phi_{VS} = -E_{IM}R^2 \sin\varphi \, (6.6R_E)^{-1} \tag{3}$$

where C is a constant equal to 92 given by Völk and Haerendel (1970), R is the geocentric distance, and $\varphi$ is the azimuthal angle. $E_{IM}$ indicates the assumed inner magnetospheric electric field derived from the solar wind electric field ($E_{SW}$), where $E_{SW}$ is computed from 1 min OMNI data (derived from upstream measurements by the Advanced Composition Explorer (ACE) spacecraft (Stone et al., 1998)). For the southward IMF, $E_{IM} = f \cdot |E_{SW}|$, where the factor $f$ is assumed to be a constant 0.13. On the other hand, in the northward IMF, $E_{IM} = f \cdot 0.25$ mV m$^{-1}$ (Goldstein et al., 2014a, b).

Based on the model of a realistic magnetospheric electric field, the evolution of the cold plasmaspheric electron distribution in the geomagnetic equator is simulated. To obtain the initial electron density distribution in the plasmasphere during the quiet geomagnetic period, the electron density in the plasmasphere as a function of the L-shell provided by the Sheeley et al. (2001) model is used (for L-shell ≤ 7), and the initial electron density is assumed to be the same at different MLTs. In addition, to simplify the calculation of the model, the electron densities outside the plasmapause are all assumed to be 5 cm⁻³. A total of 100000 test particles at an initial energy of 1 eV are launched into the model. The pitch angle of electrons is assumed to be arbitrary because the gradient/curvature drift velocity associated with the pitch angle can be negligible for cold electrons (Roederer and Zhang, 2014). The number of test particles within a unit area is transformed into a realistic density according to the weighting factor. Using the model above, the evolutions of the plasmasphere and plasmaspheric plume during different levels of geomagnetic storms are simulated. It should be pointed out that the shape of the real plasmasphere is complicated. As it is difficult to obtain the absolutely accurate shape of a real plasmasphere, a typical plasmaspheric model is used as the initial distribution of electron density in the current study. Although there may be some deviations between the simulated plume and the real plume, the simulation can still reflect the trends of density variation.

### 4.2 Plasmasphere Dynamics 13-15 February 2013

Figure 4 shows the geomagnetic and solar wind conditions for a moderate geomagnetic storm on 13-15 February 2013. As shown in Figure 4a, the minimum value of the Dst index is -37 nT during the geomagnetic storm. During the main and recovery phases of the geomagnetic storm, the IMF is southward most of the time (shown in Figure 4b). Based on the $E_{SW}$, we calculated the $E_{IM}$, which is shown in Figure 4c.

The $E_{IM}$ (derived from the $E_{SW}$) in Figure 4 was used as input to drive the test particle simulation. The simulation is started at 17:40 UT on 13 February 2013. This initial condition onset is defined as the time at which the $E_{IM}$ slope becomes positive and remains positive on its way to the maximum $E_{IM}$ value. The initial distribution of electron density is shown in Figure 5a. The





electron density is a function of the L-shells, and is provided by the model of Sheeley et al. (2001). With the dynamic evolution,
it is obvious that the plasmaspheric particles move sunward through the $E \times B$ drift within 4 hr (as shown in Figure 5b), and
the plasmapause on the nightside moves towards lower L-shells. Meanwhile, the plasmapause on the dayside temporarily
expands to higher L-shells, and its location exceeds L-shell ~8.5. Next, the solar wind-driven magnetospheric convection strips
away the outer layers of the plasmasphere. Under the combined action of convection and corotation, the plasmaspheric plume
is formed on in the afternoon side, and the location of the dayside plasmapause decreases to L-shell ~4.2 (as shown in Figure
5c). The eroded plasmaspheric material is transported sunward and may be lose to the dayside magnetopause boundary
(Spasojevic et al., 2005; Spasojevic and Inan, 2010). Meanwhile, the plasmaspheric plume is formed near the dusk side due to
the combination of convection and corotation electric fields at 20:40 UT on 14 February (as shown in Figure 5d).
To combine the simulation with the identification of plasmaspheric plumes from satellites (Cluster observations provided by
Lee et al. (2016) and Van Allen Probe observations in our study), the range of enhanced density with a specific L-shell meeting
the standard below is considered a satellite-observable plasmaspheric plume: (1) the density is more than the modeled density
of the plasmasphere provided by Sheeley et al., (2001), and (2) the isolated cycle of enhanced density with a specific L-shell
($R_{CL}$) is more than 0.2 $R_E$ but less than 2 $R_E$ (0.2 $R_E \leq R_{CL} \leq 2 R_E$). As shown in Figure 5e and f, the range of enhanced density
satisfied the criterion of an observable plasmaspheric plume from the 30th hr (23:40 UT on 14 February) to the 40th hr (09:40
UT on 15 February) at L-shell=6 (indicated by pink curve). As indicated by the black curve in Figure 5f, the Van Allen Probe
B also observed the plume from L-shell~4.7 to L-shell~5.2 at approximately 04:00 UT on 15 February 2013. There is a small
deviation between the simulated plume and the real one, which may be because the initial shape and density of real
plasmasphere is very complicated, but the real plasmasphere is hard to obtain, thus only an empirical plasmaspheric model is
adopted in the simulations. In the other intervals displayed in Figures 5c, d, g, and h, the longitudinal range of enhanced density
near L-shell=6 is too high. The wide isolated range of enhanced density near L-shell ~6 makes it difficult for the Van Allen
Probes with elliptic orbits to identify the structure as a plasmaspheric plume, because the Van Allen Probes may operate in the
high electron density region during the whole interval of the inbound and outbound orbits. Compared with that in Figure 5f,
the plasmaspheric bulge in Figures 5c, d, g and h are increasingly wider and larger, because the interplanetary magnetic field
was southward on 15 February. Although the $E_{IM}$ was small, it may have strengthened the plasmaspheric bulge near the dusk
side.
Meanwhile, as shown in Figure 5c-h, the range of enhanced density satisfied the criterion of an observable plasmaspheric
plume from the 17th hr (10:40 UT on 14 February) to the 54th hr (23:40 UT on 15 February) in at L-shell=8 (indicated by
yellow curve) during most times. Therefore, in this case of a moderate geomagnetic storm, it seems that the satellite with
higher L-shells has a larger probability of identifying the plasmaspheric plume structure than that in the inner magnetosphere
with lower L-shells.

### 4.3 Plasmasphere Dynamics 30 April -03 May 2013

Figure 6 shows the geomagnetic and solar wind conditions for a strong geomagnetic storm from 30 April to 03 May 2013. As
shown in Figure 6a, the minimum value of the Dst index is -72 nT during the geomagnetic storm. The calculated $E_{IM}$ (shown
in Figure 6c) in the main phase is much larger than that in the above moderate geomagnetic storm presented in section 4.2.
This implies that the convection during the strong geomagnetic storm was much more intense. Similar to Figure 4, the vertical
dashed line (17:00 UT on 30 April 2013) indicates that the start time of the test particle simulation corresponds to the strong
geomagnetic storm.
Figure 7 reveals the dynamic evolution of the plasmasphere and plasmaspheric plume during the strong geomagnetic storm.
The initial distribution of electron density the same as for the previous event at 17:00 UT on 30 April is shown in Figure 7a.
Due to more intense convection during the main phase of the strong geomagnetic storm, more plasmasphere material is lost.



It is obvious that the particles in the outer plasmasphere dissipate in a very short time interval, as shown in Figure 7d. The
location of the plasmapause is reduced to L-shells < 3 at 21:00 UT on 01 May (within 28 hr). Meanwhile, a typical
plasmaspheric plume structure formed near the dusk side. At 18:00 UT on 01 May 2013, the recovery phase of the geomagnetic
storm starts. As indicated by the black curve in Figure 7g, the Van Allen Probe also observed the plume from L-shell~3.4 to
L-shell~4.3 at approximately 07:00 UT on 02 May 2013. Although the $E_{IM}$ is positive in some intervals of the recovery phase,
the motions of the residual material of the plasmasphere at low L-shells (L-shell < 3) are mainly controlled by the corotation
electric field during the recovery phase. The intermittent positive $E_{IM}$ during the recovery phase of the second geomagnetic
storm may continue to bring about plume particle loss in the magnetopause, especially for the plume particles with higher L-
shells. As a result, the plasmaspheric plume becomes thinner than that during the moderate geomagnetic storm (presented in
section 4.2), especially for L-shell ≥ 8. As shown in Figures 7f-h, after 01:00 UT on 02 May, the bulged density at L-shell ~8
is too low to be identified as an observable plasmaspheric plume. Overall, the plasmaspheric plume was mainly confined to
lower L-shells (L-shell ≤ 7) in the recovery phase of the geomagnetic storm. The time interval of the Cluster satellite in the
region with L-shell ≥ 6 is much greater than that in the inner magnetosphere. As a result, during this strong geomagnetic storm,
especially the recovery phase of the geomagnetic storm, the Cluster satellite has a lower probability of identifying the
plasmaspheric plume structure than the Van Allen Probe satellites (in the inner magnetosphere with lower L-shells).
**5 Discussion and Conclusion**
In the present study, using density data from Van Allen Probe A, we performed a statistical analysis of plasmaspheric plumes
in the inner magnetosphere. A total of 422 plasmaspheric plume events are captured out from 4030 Van Allen Probe A orbits.
The statistical results show that the ratio of observed plasmaspheric plume events is largest (~43%) during the nonstorm period.
This may be because the plasmaspheric plume that forms during a geomagnetic storm, may remain residual for quite a long
time period after the geomagnetic activity has recovered. In addition, quiet geomagnetic activity occupies most of the time
interval (Halford et al., 2010; Wang et al., 2016). Therefore, the number of observed plasmaspheric plume events during the
nonstorm period is high. Since the corotation electric field plays a leading role in the motion of plasmaspheric particles during
quiet geomagnetic activity, the residual plasmaspheric plume can corotate with the Earth. Consequently, the residual
plasmaspheric plume may be observed by satellite in all MLTs (as shown in Figure 2a).
Moreover, during the interval of geomagnetic storms, plasmaspheric plume events are mainly concentrated in the recovery
phase and dusk side. This result is similar to the conclusions of previous works, such as Chi et al. (2000), Reinisch et al. (2004),
and Kim et al. (2007), and suggests that the structure of the plasmaspheric plume appears is more obvious after the large
erosion in the main phase of geomagnetic storms. However, this result is different from the observation at the magnetopause.
Walsh et al. (2013) suggested that the most common location where plume material contacts the magnetopause is at MLT~13.6.
This may be because the plasma material is dragged from the dusk region with lower L-shells towards the noon side with
higher L-shells due to sunward convection.
In this study, to investigate the correlation between the occurrence rate of observed plasmaspheric plumes in the inner
magnetosphere and the level of geomagnetic storms, we select the minimum Dst value from the previous 24 hr to account for
the response time of the plasmapause to geomagnetic storms. The results show that the occurrence rate of observed
plasmaspheric plumes in the inner magnetosphere increases with increasing geomagnetic activity, and the largest occurrence
rate corresponds to the most intense geomagnetic activity. This result is different from the occurrence rate of observed
plasmaspheric plume events detected by the Cluster satellite with a much higher apogee, which was presented in Lee et al.
(2016). They suggested that the plasmaspheric plume events observed by the Cluster satellite tend to be observed during
moderate geomagnetic activity. The dynamic evolutions of the plasmaspheric plume are simulated during both moderate and
strong geomagnetic storms to demonstrate the disparity of observations at different L-shells. The simulation results suggest

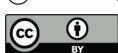



that plasmasphere erosion is smaller and that the range of plasmaspheric plumes in the inner magnetosphere is wider during
moderate geomagnetic activity (as shown in section 4.2). The wider isolated region of high density contributed by
plasmaspheric plumes near L-shells ≤ 6.2 may make it difficult for the Van Allen Probes with elliptic orbits to identify the
structure as an observed plasmaspheric plume. In addittion, the isolated region of high density contributed by plasmaspheric
plumes is narrower when L-shells ≥ 8, which make it easy for Cluster (with higher L-shells during most of the orbital period)
to identify the plasmaspheric plume structure during moderate geomagnetic storms, especially in the recovery phase. It must
be admitted that the magnetic field is assumed to be a dipolar field in this study, so the calculations of electron motions are not
entirely correct near the magnetopause. Nonetheless, it can generally reflect the trend of electron density within L-shells ≤ 8.5,
which is exhibited in Figures 5 and 7.
On the other hand, the simulated scale of plasmaspheric plumes during strong geomagnetic storms is different from that during
moderate geomagnetic storms. As presented in section 4.3, plasmasphere erosion is extremely intense during the main phase
of a strong geomagnetic storm. A great quantity of outer plasmaspheric particles is lost outside the magnetopause. The
plasmapause shrank to L-shells < 3 when the recovery phase started, and the residual plasmasphere may be primarily controlled
by the corotation electric field. During the recovery phase of strong geomagnetic storm, the plasmaspheric plume is much
thinner and narrower than the plasmaspheric plume during a moderate geomagnetic storm. Consequently, the Van Allen Probes
more easily identify the structure of plasmaspheric plumes during the recovery phase of strong geomagnetic storms. In addition,
the enhanced density near the magnetopause contributed by the stretched plasmaspheric plume is too low during strong
geomagnetic storms. The obvious structure of the plasmaspheric plume is confined to lower L-shells. As a result, the Cluster
satellites with higher L-shells in most orbital periods have difficulty identifying the structure of plasmaspheric plumes during
strong geomagnetic storms.
In summary, the main conclusions of the study are as follows:
1. The plasmaspheric plume events during the nonstorm period are distributed in all MLTs, but the number of plasmaspheric
plume events from the dusk side to the midnight side is the largest. In addition, during geomagnetic storms, the plasmaspheric
plume events tend to occur near the dusk side during the recovery phase.
2. The plasmaspheric plume in the inner magnetosphere is preferentially observed during strong geomagnetic storms. This
result is different from the statistical results of observations near the magnetopause, which suggest that the plasmaspheric
plume tends to be observed during moderate geomagnetic storms.
3. The evolutions of plasmaspheric plumes during moderate and strong geomagnetic storms were simulated, respectively.
During the case of the moderate geomagnetic storm, the wider isolated region of high density contributed by the plume may
make it difficult for the Van Allen Probes in the inner magnetosphere to identify the structure as an observed plasmaspheric
plume. However, the region of high density contributed by the plasmaspheric plume is narrower near the magnetopause, which
makes it easy for the satellite near magnetopause to identify the plasmaspheric plume structure.
4. During the case of the strong geomagnetic storm, the plasmapause shrank to a very low L-shell, and the scale of the plume
was narrower, and these two results in the Van Allen Probes in the inner magnetosphere frequently identify the structure of
the plasmaspheric plume. In addition, the plasmaspheric plume may be confined to lower L-shells, which makes it difficult for
the Cluster satellite to identify the plasmaspheric plume structure.
Notably, the cases above cannot represent all the evolutions of plasmaspheric plumes during either moderate or strong
geomagnetic storm. However, this study provides an alternative mechanism to interpret the different occurrence rates of
plasmaspheric plumes detected by different satellites. Furthermore, since a relatively long time is required for the plasmasphere
to recover to a normal level after a geomagnetic storm (Xiao-Ting et al., 1988; Chu et al., 2017), we did not consider the
refilling process of the plasmasphere from the ionospheric particles drawn upward.
More theoretical and comprehensive modeling will be studied in our future project.



*Data availability.* The data of EMFISIS aboard Van Allen Probes are download from http://emfisis.physics.uiowa.edu/Flight/.
The data of OMNI are from  http://cdaweb.gsfc.nasa.gov.
*Author contributions.* The conceptional idea of this study was developed by HL and RT. HL and TF wrote the paper, and RT
revised it. ZY and XD substantially helped with the analysis. OZ contributed to the Van Allen Probe data processing. All
authors discussed the results.
*Competing interests.* The authors declare that they have no conflict of interest.
*Acknowledgements.* This research is supported by the National Natural Science Foundation of China (Nos. 42064009,
41974195, 41674144). The data of EMFISIS are from http://www.space.umn.edu/rbspefw-data/. The *DST* data are provided
by OMNI at http://cdaweb.gsfc.nasa.gov.

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

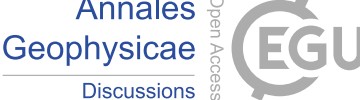

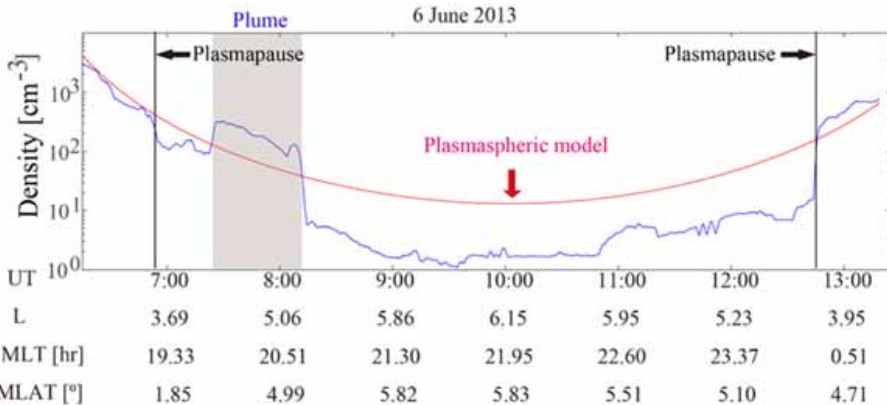

**Figure 1.** A typical example of a plasmaspheric plume measured by Level 4 EMFISIS data sets of Van Allen Probe A. The measured electron density and the density provided by Sheeley et al. (2001) are indicated by blue and red curves, respectively. The black vertical lines denote the location of the plasmapause as determined by Moldwin et al. (2002). The gray shadow indicates the region of the detected plasmaspheric plume.



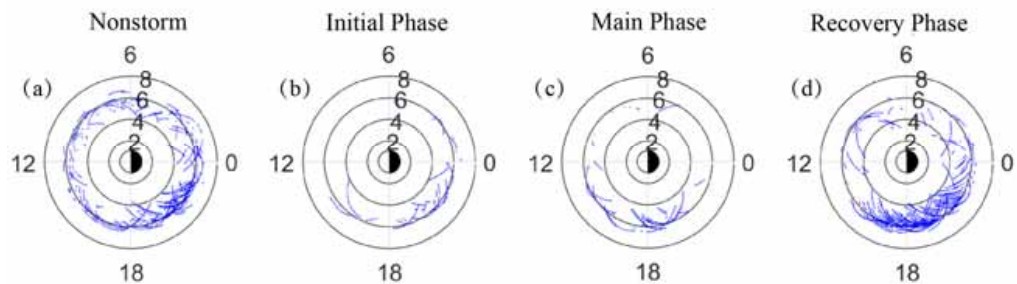

**Figure 2.** The spatial distribution of plasmaspheric plumes (422 total events from January 2013 to December 2018) are shown in the MLT-*L* plane. (a–d) The distributions of observed plasmaspheric plumes during the nonstorm period (185 events), initial phase (31 events), main phase (32 events), and recovery phase (174 events).





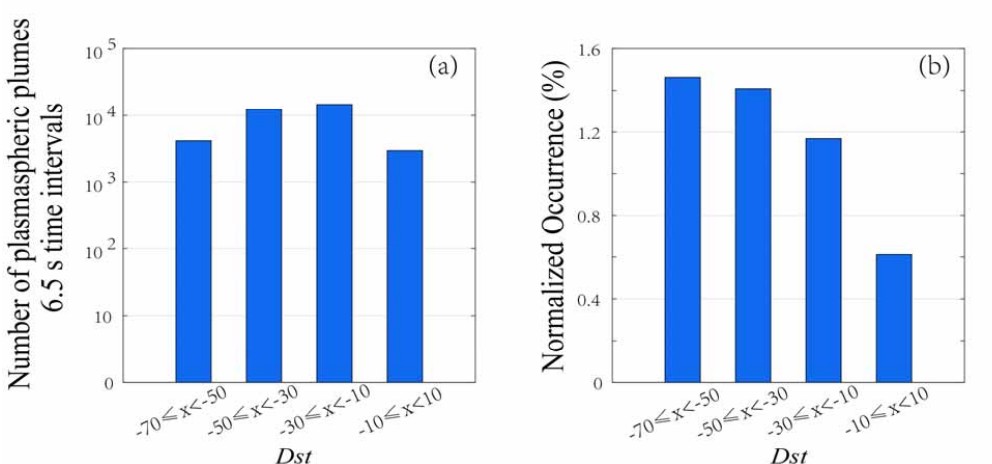

419

**Figure 3.** (a) The distribution of observed plasmaspheric plume density data points as a function of the minimum *Dst* in the

previous 24 hr. (b) The normalized occurrence rates of plasmaspheric plumes in the inner magnetosphere with respect to the

minimum *Dst* in the previous 24 hr.





424

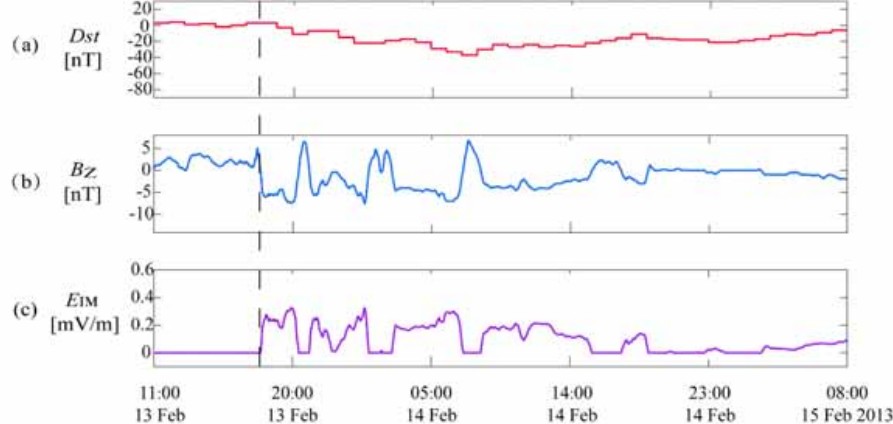

425

**Figure 4.** Geomagnetic and solar wind conditions on 13-15 February 2013. The vertical dotted line indicates the start time of the test particle simulation (17:40 UT on 13 February 2013). (a) *Dst* index. (b) z component of IMF in GSM coordinates from merged 1 min OMNI data. (c) Assumed inner magnetospheric $E_{IM}$ derived from $E_{SW}$ (see text).

429



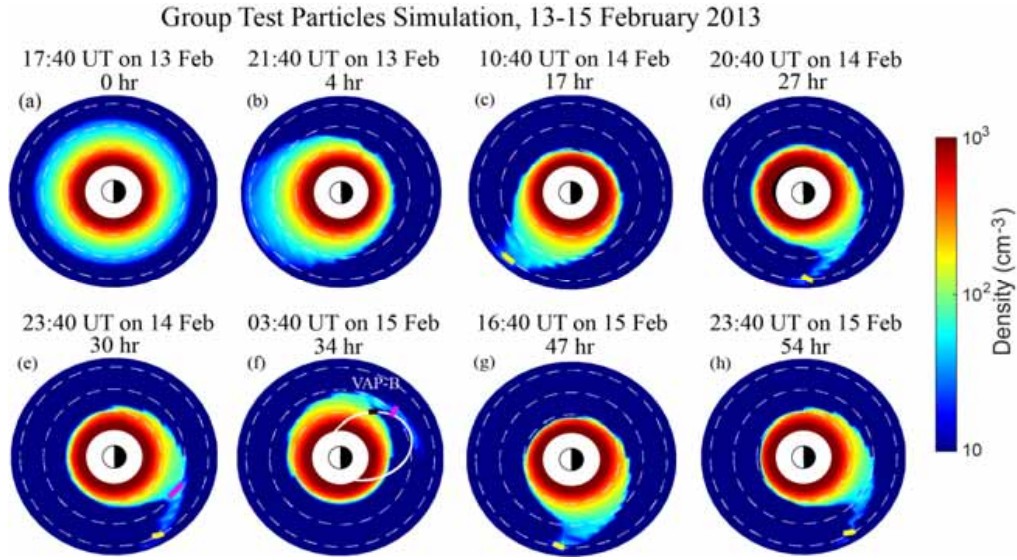

430

431

**Figure 5.** The equatorial plots of the simulated plasmasphere and plasmaspheric plume through test particle simulation during 13-15 February 2013. The white curve represents the orbit of the Van Allen Probe B satellite from 22:00 UT on 14 February to 07:00 on 15 February 2013. The black curves indicate the observed plasmaspheric plume. The white dashed circles represent $L$-shells =4, 6, and 8. The time above each panel represents the evolution time of the plasmasphere and plasmaspheric plume. The pink (yellow) curve indicates the range of enhanced density with a specific $L$-shell =6 ($L$-shell =8) that meets the standard of a satellite-observable plasmaspheric plume.

438



439

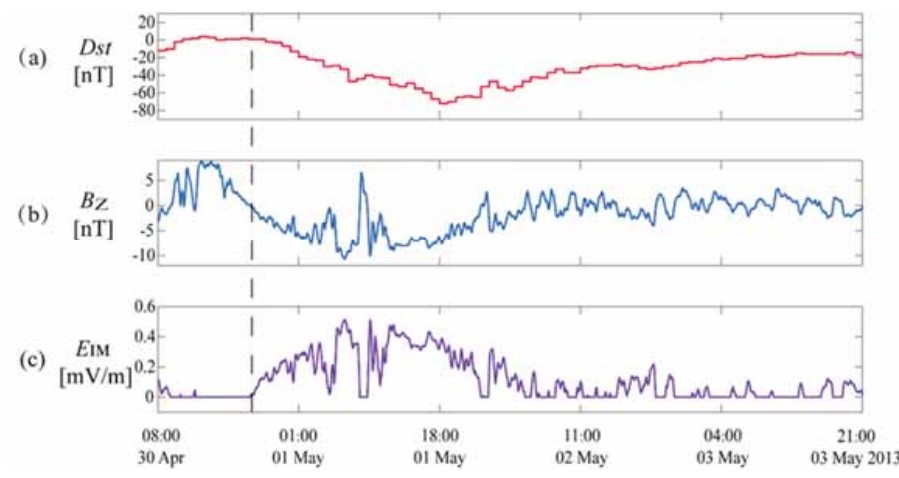

440

441

442   **Figure 6.** Geomagnetic and solar wind conditions on 30 April-03 May 2013. The format is the same as Figure 4.

443



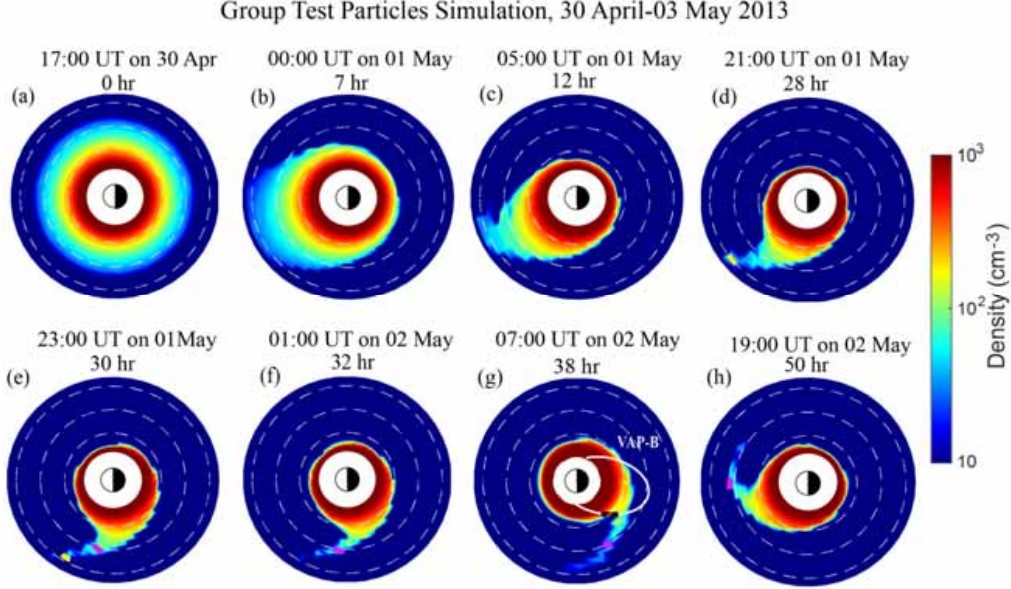

**Figure 7.** The equatorial plots of the simulated plasmasphere and plasmaspheric plume through test particle simulation during 30 April-03 May 2013. The white curve represents the orbit of the Van Allen Probe B satellite from 06:00 UT to 15:00 on 02 May 2013. The black curves indicate the observed plasmaspheric plume. The white dotted circles represent $L=4$, 6, and 8. The number on each plot represents the time of evolution. The pink (yellow) curve indicates the range of enhanced density with a specific $L=6$ ($L=8$) that meets the standard of a satellite-observable plasmaspheric plume.