# Peer review of "Statistical Study and Corresponding Evolution of Plasmaspheric Plumes under Different Levels of Geomagnetic Storms"

_Annales Geophysicae, 2021_

## Author Comment (AC1)

**Response to the Reviewer #1**

Dear Editor and Reviewers:

Many thanks for your careful reading and good comments for our manuscript entitled " Statistical Study and Corresponding Evolution of Plasmaspheric Plumes under Different Levels of Geomagnetic Storms". These valuable suggestions are really beneficial to our present research. We've carefully considered these suggestions and implemented the necessary changes, which we hope will adequately address the reviewers' concerns. The following are our primary replies to the reviewer's comments and the corresponding corrections.

**Responses to the Reviewer 1:**

**Comments from Reviewer 1:**

The preprint is dealing with plasmaspheric plumes in the Earth's magnetosphere and their occurrence rate dependence to geomagnetic storms intensity. The study is based on observations and simulations. The study is based on Van Allen Probe A observations of plasmaspheric plumes. Results are compared in detailed to the output of a previous study (Lee et al., 2016) that focused on the observations of plumes close to the magnetopause. Simulations of plume evolution are presented for two geomagnetic storms of different intensities to explain the different results between the present study and Lee et al. study. The overall impression of the study is positive: the paper is well-organized, the reasoning is logical, figures support the interpretation, and considering simulations to explain observations make the conclusion stronger.

Specific comments:

1. Only VAP-A observations are used (and not VAP-B), which make sense for the occurrence rate calculations. It is therefore surprising that only VAP-B orbit is overplotted on the simulation output with the indication "the black curves indicate the observed plasmaspheric plume" (Figure 5 and Figure 7). Why is that plume observation made by VAP-B? one could expect observations from VAP-A or, even better from both probes.

**Answer:** Thank you very much for your suggestion. As you kindly suggest, only VAP-B orbit is overplotted on the simulation output. The simulation output largely depends on the initial configuration of plasmasphere. The initial shape and density of real plasmasphere are very complicated, and there is obvious asymmetry of Earth's plasmasphere. However, the real initial configuration of plasmasphere is hard to

obtain, thus only an empirical plasmaspheric model is adopted in the simulations. The output implies that there is larger deviation between the plume MLT observed by VAP-A and that provided by simulation. On the other hand, the deviation between the plume MLT observed by VAP-B and that provided by simulation is smaller. For those reasons, we only depict the orbit of VAP-B in the study.

However, the simulation in the study provides an alternative mechanism to interpret the evolution trend and occurrence rates of observable plasmaspheric plumes in different L shells.

In order to make it clearer, the sentences have been added as follows:

On lines 192-193:

'However, the real plasmasphere is hard to obtain, thus only an empirical plasmaspheric model is adopted in the simulations.'

On lines 160-164:

'It should be pointed out that the shape of the real plasmasphere is complicated. As it is difficult to obtain the absolutely accurate shape of a real plasmasphere, a typical plasmaspheric model is used as the initial distribution of electron density in the current study. Although there may be some deviations between the simulated plume and the real plume, the simulation can still reflect the trends of density variation.'

2. The main finding of the study is that the occurrence rate of plumes at lower l-shell is higher during large geomagnetic storms and higher at large l-shell during low geomagnetic storms. As shown in your simulations a given plume plume is larger close to the plasmapause (at low-shell)  that farther from the plasmapause (at larger L-shell). The plume criteria detection is however the same for every L-shells. How can this affect the occurrence rate estimation, and hence the conclusion?

**Answer:** Thank you very much for your suggestion. As the reviewer kindly suggest, the criteria detection is the same for every L-shells, this may affect the estimation of occurrence rate. In fact, in previous studies, the researchers generally using the same criteria to identify the observable plume in different L shells, such as Lee et al., (2015), Zhang et al., (2019).

The plasmaspheric plume is structure detached plasma region which connect to main body of the plasmasphere. A key point of the study is to explain the proportion of observable plasmaspheric plume in different L shells associated with geomagnetic activity on account of traditional criteria.

In order to explain better, the sentences have been revised as follows:

On lines 256-258:

'The researchers generally using the same criteria to identify the observable plume in different L shells, such as Lee et al., (2015), Zhang et al., (2019). A key point of the study is to explain the proportion of observable plasmaspheric plume in different L-shells associated with geomagnetic activity.'

3. The two simulated events correspond to geomagnetic storm of still relatively moderate intensity. Is there any reason for not choosing a Dst<-100nT storm for the second event?

**Answer:** Thank you very much for your suggestion. Through the statistical analysis based on the observation of Van Allen Probe, we find that the number of plume events corresponding to Dst<-100 nT is very few. Only 2 observable plasmaspheric plume events is located in this Dst range (the total number of plasmaspheric plume is 422). In order to reflect the universality of evolutionary process the extreme cases while Dst<-100nT is not choosed for the second event.

4. L45: for EMIC the situation is a bit more complicated: EMIC are not preferentially observed in the high-density plumes (Usanova) excepted maybe for triggered emission (Grison)

**Answer:** Thank you very much for your reminder. In order to make it clearer, the sentence has been revised as follows:

On lines 45-49:

'The structureless hiss waves often arise in high-density plasmaspheric plumes (Meredith et al., 2004; Yuan et al., 2012; Zhang et al., 2018, 2019). Furthermore, although electromagnetic ion cyclotron (EMIC) waves are not preferentially observed in the high-density plumes (Usanova et al., 2013; Grison et al., 2018), the plume maybe related to the excitation of EMIC waves (Grison et al., 2018; Yu et al., 2016; Yuan et al., 2010).'

5. L146: Is there any discontinuity in the density or the Sheeley model states a density of 5 at L=7?

**Answer:** Thank you very much for your suggestion. Sheeley model is only valid for the region within L≤7. In the trough model from Sheeley et al. (2001), the density is ~5 $cm^{-3}$. To clearly exhibit the position of plasmapause for the simulation. Here, the electron densities outside the plasmapause are assumed to be 5 $cm^{-3}$, the electron

density while L ≤ 7 is provided by the Sheeley plasmaspheric model. In this way, a relatively high density gradient (the position of plasmapasue) can be setted around L~7. In fact, the result of simulation is similar while the density is set to any value less than 5 cm$^{-3}$ in the region with L>7.

In order to make it clearer, the sentences have been revised as follows:

On lines 151-156:

'To obtain the initial electron density distribution in the plasmasphere during the quiet geomagnetic period, the electron density in the plasmasphere as a function of the L-shell provided by the Sheeley et al. (2001) model is used (for L-shell ≤ 7). In order to clearly exhibit the position near the plasmapause, the initial electron density is assumed to be the same at different MLTs. In addition, to simplify the calculation of the model, the electron densities outside the plasmapause are all assumed to be 5 cm$^{-3}$. In this way, a relatively high density gradient is assumed around L~7.'

6. Figure 4.c: It would be good to extend the time range to the stop time of the simulation (figure 5)

**Answer:** Thank you very much for your suggestion. In the rewvised manuscript, the end time of Figure 4 is set at 00:00 UT on 16 Feb 2013.

7. Technical corrections:

L25: A torus

**Answer:** Thank you very much for your suggestion, the error has been revised in the new version of manuscript.

L30: plasmaspheric particles: the convection given by the formula is sign-charge dependent

**Answer:** The directions of both gradient and curvature drifts depends on the sign-charge. On the other hand, the drift motion is independent of the sign charge. The velocity of E×B drift is: $\frac{E \times B}{B^2}$

L72: deleted artificially: "discarded" might sound better

**Answer:** Thank you very much for your suggestion. The sentence has been revised as follow: '...these spurious events are discarded...'

L94: define the "non storm" period

**Answer:** Thank you very much for your suggestion. In order to make it clearer, the sentence has been revised as follows:

'The period except strom time interval (including initial, main and recovery phases) is defined as nonstorm.'

L101: dusk side: looking at the figure, you could even give a precise range 15-21MLT

**Answer:** Thank you very much for your suggestion. The sentence has been revised as follow:

On lines 107-108:

'it seems that the plasmaspheric plume events observed in the recovery phase (174) occupy the largest proportion, and the plasmaspheric plumes in the recovery phase are mainly located on the dusk side (from MLT~15 to MLT~21).'

---

## Author Comment (AC2)

**Response to the Reviewer #2**

Dear Editor and Reviewers:

Many thanks for your careful reading and good comments for our manuscript entitled " Statistical Study and Corresponding Evolution of Plasmaspheric Plumes under Different Levels of Geomagnetic Storms". These valuable suggestions are really beneficial to our present research. We've carefully considered these suggestions and implemented the necessary changes, which we hope will adequately address the reviewers' concerns. The following are our primary replies to the reviewer's comments and the corresponding corrections.

**Responses to the Reviewer 2:**

**Comments from Reviewer 2:**

Plasmaspheric plumes play important roles in the inner magnetosphere. The current study statisticaly investigats the plumes observed by Van Allen Probes. Further more, it explains the difference of the observed features of plumes by Van Allen Probes and Cluster by performing test particle simulations. The simulations results explain well these different features. This is an interesting study and contributs to our understanding about plasmaspheric plumes. I suggest this preprint be published after minor revisions and list my comments and suggestions as follows:

Specific comments:

1. A general question:during these intervals under study, where were Cluster satellites? Since authors compare observation results from Cluster and Van Allen Probes, is it possible to compare their observations during the same time intervals?

**Answer:** Thank you very much for your suggestion. Since the Cluster satellite don't directly provide the background electron density. The plume is often identified by the dift ion flux observed by the Cluster satellite (it is some complicated to set a standard). Furthermore, in the study, a dipolar magnetic field is adopted in the simulation, it may not be entirely suitable for the region of Cluster satellite operate with higher L shells. As a result, the observation of Cluster isn't exhibited in the study. A developed model of simulation adopting Tsyganenko magnetic field will be studied in the future.

However, the simulation in the study provides an alternative mechanism to interpret the evolution trend and occurrence rates of observable plasmaspheric plumes in different L shells.

In order to make it clearer, the sentence has been added as follows:

On lines 266-267:

' It must be admitted that the magnetic field is assumed to be a dipolar field in this study, so the calculations of electron motions are not entirely correct near the magnetopause.'

2. Line 55: 'In this paper', in situ measurements from Van Allen Probes are used to…

**Answer:** Thank you very much for your suggestion. The sentence has been revised as follow:

' in situ measurements from Van Allen Probes are used to'

3. Line 73: what is the criteria for this 'sharply'?

**Answer:** Thank you very much for your suggestion. The sentence has been revised as follow: (On lines 75-76)

'...where the observed electron density sharply increases (by a factor >5 within 0.5 L-shell)...'

4. Line 102: But the time interval of them are shorter. Could you please normalize them and compare the occurance rate rather than simple number of events?

**Answer:** Thank you very much for your suggestion. As you suggest, the interval of main phase is shorter, on the other hand, both the interval of recovery phase and quiet time are much longer.

At present, the calculation of occurrence rate during different phase may be a very complex task in a short term. In the study, in order to ensure the accuracy of the judgement, the corresponding phases of plume events are distinguished through manual identification. If following this method, we need to determine the duration of each phase in each geomagnetic storm from 2013 to 2018, this is a very big job. As a result, the study mainly focus on the number of events.

Of course, it is very important to calculate the occurrence rate during different phase of geomagnetic storm, maybe we will do it in the future.

The explanation of the distribution has been described in the section of 'Discussion and Conclusion':

On lines 236-241:

'This may be because the plasmaspheric plume that forms during a geomagnetic

storm, may remain residual for quite a long time period after the geomagnetic activity has recovered. In addition, quiet geomagnetic activity occupies most of the time interval (Halford et al., 2010; Wang et al., 2016). Therefore, the number of observed plasmaspheric plume events during the nonstorm period is high. Since the corotation electric field plays a leading role in the motion of plasmaspheric particles during quiet geomagnetic activity, the residual plasmaspheric plume can corotate with the Earth. Consequently, the residual plasmaspheric plume may be observed by satellite in all MLTs (as shown in Figure 2a).'

5. Line 114: It would be interesting to add MLT-L dependence. Maybe authors can plot 'dial' figures like in Figure 7, but color-code occurance rate in different MLT-L bins.

**Answer:** Thank you very much for your suggestion. As shown in Figure 1, the widths of plasmaspheric plume are very variable, some cases of plasmaspheric plume span several MLTs, and the ranges for some special cases are smaller. As a result, we plot the spatial distribution of plasmaspheric plumes based on the trajectory detected by the satellite in Figure 1. In addition, the apogee of VAP satellites encircle the earth once every ~18 months. In the study, the data observed by VAP-A from January 2013 to December 2018 are adopted. This ensure that the distributions of sampling points on MLTs are uniform (four full cycles of VAP apogee).

6. Line 115: How about storms with Dst lower than -70 nT?

**Answer:** Thank you very much for your suggestion. The number of plasmaspheric plume with Dst lower than -70 nT is very few, the number is only 6. The number is too small to compare with the other range. Furthermore, in the study of Lee et al., (2015) based on the observation of Cluster satellite, the corresponding range is from Dst~-50 to Dst~10. For the above reasons, the Dst is limited above -70 nT in the study.

In order to make it clearer, the sentence has been added as follow:

On lines 113-114:

'Similar to the analysis of the relationship between the plasmaspheric plume near magnetopause and geomagnetic activity studied in Lee et al. (2016) (which reveal statistical analysis of plumes while Dst >-50 nT)...'

7. Line 125: delete 'disparity' after 'rates'

**Answer:** Thank you very much for pointing out our mistake, we have revised the sentences as follows:

On lines 132-133:

'To explain the disparity in the occurrence rates of the observed plasmaspheric plume associated...'

8. Line 127: test particle simulation's'

**Answer:** Thank you very much for your suggestion, we have revised it in the new version of manuscript.

9. Line 146: is the 5cc set up for the initial condition?

**Answer:** Thank you very much for your suggestion. In the study, in order to clear exhibit the position of plasmapause (steep gradient of electron density), the 5cc is set up for the initial condition.

In order to make it clearer, the sentence has been revised as follow:

On lines 151-156:

'To obtain the initial electron density distribution in the plasmasphere during the quiet geomagnetic period, the electron density in the plasmasphere as a function of the L-shell provided by the Sheeley et al. (2001) model is used (for L-shell ≤ 7). In order to clearly exhibit the position near the plasmapause, the initial electron density is assumed to be the same at different MLTs. In addition, to simplify the calculation of the model, the electron densities outside the plasmapause are all assumed to be 5 cm$^{-3}$. In this way, a relatively high density gradient is assumed around L~7.'

10. Line 166: I suggest add labels indicating some L values in the figure

**Answer:** Thank you very much for your suggestion. We have added labels indicating L~4, L~6, L~8 in the first panels of Figure 5 and Figure 7 .

11. Line 168: remove 'in' after 'on'

**Answer:** Thank you very much for your suggestion, we have revised in the new version of manuscript.

12. Line 169: be lose → lost

**Answer:** Thank you very much for your suggestion, we have revised in the new version of manuscript.

13. Line 177: 40th hr (not shown)

**Answer:** Thank you very much for your suggestion. Since around 38$^{th}$ hr, the VAP observes the structure of plasmaspheric, the corresponding simulation result around 38$^{th}$ hr is displayed. The two moments (38th and 40th) are too close. As a result, the simulation at 40$^{th}$ hr isn't shown in the Figure.

In order to make it clearer, the sentence has been revised as follows:

On lines 188-189:

'...from the 30th hr (23:40 UT on 14 February) to the 40th hr (09:40 UT on 15 February, not shown here)...'

14. Lines 179-182: This sentence is too long. I suggest authors to finish a sentence after 'complicated' on line 181 and to start a new sentence afterwards.

**Answer:** Thank you very much for your suggestion. The sentences have been revised as follows:

On lines 191-193:

'There is a small deviation between the simulated plume and the real one, which may be because the initial shape and density of real plasmasphere is very complicated. However, the real plasmasphere is hard to obtain, thus only an empirical plasmaspheric model is adopted in the simulations.'

15. Line 203: 'The initial distribution of electron density' is set up in the same way as … on 30 April, and is shown …

**Answer:** Thank you very much for your suggestion. The sentences have been revised as follows:

On lines 214-215:

'The initial distribution of electron density is set up in the same way as the previous event at 17:00 UT on 30 April, and is shown in Figure 7a.'

16. Line 212: loss 'to' the magnetopause

**Answer:** Thank you very much for your suggestion. The sentence has been revised in the new version of manuscript.

17. Line 213: Is the upflow of electrons also stronger in strong storms? This can be an uncertainty in your simulation studies since you don't have the upflow process included in your simulations but they can be different for storms of different levels.

**Answer:** As the reviewer kindly suggest, after disturbance is subsided, the electric field recovers and the plasmasphere starts refilling from the ionospheric plasma due to upflow of electrons. This may be an uncertainty in the simulation.

In order to make it clearer, the sentences have been revised as follows:

On lines 296-301:

'Notably, the cases above cannot represent all the evolutions of plasmaspheric plumes during either moderate or strong geomagnetic storm. However, this study provides an alternative mechanism to interpret the different occurrence rates of plasmaspheric plumes detected by different satellites. Furthermore, since a relatively long time is required for the plasmasphere to recover to a normal level after a geomagnetic storm (Xiao-Ting et al., 1988; Chu et al., 2017), we did not consider the refilling process of the plasmasphere from the ionospheric particles drawn upward. More theoretical and comprehensive modeling will be studied in our future project.'

18. Line 226: Again, I suggest calculating occurrance rates, instead of simply comparing number of events.

**Answer:** Thank you very much for your suggestion. As you suggest, the interval of main phase is shorter, on the other hand, both the interval of recovery phase and quiet time are much longer.

At present, the calculation of occurrence rate during different phase may be a very complex task in a short term. In the study, in order to ensure the accuracy of the judgement, the corresponding phases of plume events are distinguished through manual identification. If following this method, we need to determine the duration of each phase in each geomagnetic storm from 2013 to 2018, this is a very big job. As a result, the study mainly focus on the number of events.

Of course, it is very important to calculate the occurrence rate during different phase of geomagnetic storm, maybe we will do it in the future.

19. Line 232: remove 'appears' after 'plume'

**Answer:** Thank you very much for your suggestion. The 'appears' has been removed in the new version of manuscript.

20. Line 257: lost 'to' the magnetopause

**Answer:** Thank you very much for your suggestion. The sentences have been revised as

On line 271

'...plasmaspheric particles is lost to the magnetopause...'.

21. Lines 279: 'these two' factors make 'the Van Allen Probes'

**Answer:** Thank you very much for your suggestion. The sentences have been revised as: '...and these two factors make the Van Allen Probes in the inner magnetosphere...'.

22. Figure 1: Considering this is a study related with storm/non-storm periods, I suggest authors to add panels in this figure to show related geomagnetic indices (e.g., Dst, Kp, AU, AL, AE), and to add verticl lines indicating storm phases and the start of the storm if this is a storm period.

**Answer:** Thank you very much for your reasonable suggestion. We have added the related geomagnetic indices (Dst, Kp) in the new version of manuscript. And the sentences have been added in the new version of manuscript:

On lines 83-85:

'Figure 1 displays an example of a plasmaspheric plume observed by Van Allen Probe A on 6 June 2013. The top panels exhibit the geomagnetic indices (including Dst and Kp) from 6 June to 7 June in 2013. The measured density from 06:35 UT to 14:00 UT is shown in the bottom panel.'

On lines 89:

'We find that the plume occurs in the non-storm period through the analysis of geomagnetic indices.'

On lines 420-425:

'**Figure 1.** A typical example of a plasmaspheric plume measured by Level 4 EMFISIS data sets of Van Allen Probe A. The top panels exhibit the geomagnetic indices (including Dst and Kp) from 00:00 UT on 6 June to 00:00 UT on 7 June in 2013. The measured density from 06:35 UT to 14:00 UT is shown in the bottom panel. The measured electron density and the density provided by Sheeley et al. (2001) are indicated by blue and red curves, respectively. The black vertical lines denote the location of the plasmapause as determined by Moldwin et al. (2002). The brown shadows indicate the time interval of the detected plasmaspheric plume.'

23. Figure 7 caption: line 449: on → above

**Answer:** Thank you very much for your suggestion. The sentence has been revised as follow:

'The number above each plot represents the time of evolution.'